# High-Throughput Separation of Long DNA in Deterministic Lateral Displacement Arrays

**DOI:** 10.3390/mi13101754

**Published:** 2022-10-17

**Authors:** Oskar E. Ström, Jason P. Beech, Jonas O. Tegenfeldt

**Affiliations:** Division of Solid State Physics, Department of Physics and NanoLund, Lund University, P.O. Box 118, 22100 Lund, Sweden

**Keywords:** microfluidic separation, deterministic lateral displacement, long DNA sample preparation, high throughput, concentration effects

## Abstract

Length-based separation of DNA remains as relevant today as when gel electrophoresis was introduced almost 100 years ago. While new, long-read genomics technologies have revolutionised accessibility to powerful genomic data, the preparation of samples has not proceeded at the same pace, with sample preparation often constituting a considerable bottleneck, both in time and difficulty. Microfluidics holds great potential for automated, sample-to-answer analysis via the integration of preparatory and analytical steps, but for this to be fully realised, more versatile, powerful and integrable unit operations, such as separation, are essential. We demonstrate the displacement and separation of DNA with a throughput that is one to five orders of magnitude greater than other microfluidic techniques. Using a device with a small footprint (23 mm × 0.5 mm), and with feature sizes in the micrometre range, it is considerably easier to fabricate than parallelized nano-array-based approaches. We show the separation of 48.5 kbp and 166 kbp DNA strands achieving a significantly improved throughput of 760 ng/h, compared to previous work and the separation of low concentrations of 48.5 kbp DNA molecules from a massive background of sub 10 kbp fragments. We show that the extension of DNA molecules at high flow velocities, generally believed to make the length-based separation of long DNA difficult, does not place the ultimate limitation on our method. Instead, we explore the effects of polymer rotations and intermolecular interactions at extremely high DNA concentrations and postulate that these may have both negative and positive influences on the separation depending on the detailed experimental conditions.

## 1. Introduction

There is renewed interest in fractionation of long DNA molecules due to the development of sequencing techniques that support long read lengths, notable single-molecule real-time (SMRT) sequencing from PacBio and nanopore systems from Oxford Nanopore Technology. “Long” DNA refers to the upper length limit of standard gel electrophoresis, approximately 20 kbp. For sequencing applications, long read lengths enable detection of large structural variations [1], and open up the analysis of plant genomes that has been limited by current technology [2].

The SMRT system from PacBio is based on a polymerase that is located in a zero-mode waveguide where it feeds the DNA while fluorescently labelled bases are detected as they are incorporated. The read length depends on the lifetime of the polymerase and is typically 5–10 kbp [3]. The nanopore technology, on the other hand, does not have that type of inherent limit on maximum read length. Instead, the purity and the composition of the sample determine the resulting read length. The longest reported read length that we could find in the literature was 2.3 Mbp [4], but for standard sequencing applications, 10–30 kbp is typically reached [5].

It is, therefore, essential to maximize the proportion of the DNA of interest in the sample that will be sequenced. Conventional approaches, such as a modern variant of Pulsed-Field Gel Electrophoresis (PFGE), e.g., BluePippin from Sage Science, work well but take significant time to process. For example, it takes up to ten hours for a 50 kbp size selection [6].

Instead, microfluidic alternatives have been demonstrated to offer significantly faster preparations of the DNA samples. The first microfluidic alternative to PFGE was initially introduced by Volkmuth and Austin [7]. It was further refined in a device consisting of a hexagonal array where the DNA was moved electrophoretically by a field pulsing in two directions [8] to accomplish the separation of Bacterial Artificial Chromosomes (BACs) within 15 s. Another approach was based on entropic traps and Ogsten sieving, performing a similar sorting in 15 min [9]. Deterministic lateral displacement (DLD) is another approach which has been demonstrated as being able to sort the DNA quickly. DLD relies on the repeated bifurcation of flow streams around obstacles in a microfluidic array [10]. The result is that, while small particles follow the flow, larger particles are laterally displaced through the interaction with the pillars and can be collected downstream in a separate reservoir. The main advantage of these microfluidic approaches is their high speed and their amenability to continuous and automated operation.

However, an important general limitation of microfluidics is its low throughput (see Appendix A for an overview of devices for DNA separation). For example, DLD relies on the steric displacement of particles with finite size. At very low shear rates, the DNA maintains a blob-like conformation, thereby behaving as a particle with a radius on the order of its radius of gyration that, in turn, is dependent on its length. At higher shear rates, the DNA is stretched out, appearing as a thin rod with a diameter that is independent of its length and this is generally understood to prevent any sorting [7,10,11]. One approach to increase throughput at low shear rates is to couple many devices in parallel [11,12], but this is costly and requires that the fluidics through all devices are carefully matched. Another alternative is to suppress the extension of the DNA in the shear flow. This has been done using depletion forces with a carefully tuned concentration of polyethylene glycol (PEG) [13] that condenses the DNA, effectively increasing its shear modulus.

In this work, we will address the limitation in throughput by a simple method based on DLD separation of high concentrations of the DNA at high flow velocities. We show that it is possible to displace long DNA molecules despite their considerable extension at these flow velocities and propose some possible mechanisms for our findings.

## 2. Materials and Methods

### 2.1. Device Design

In DLD, particles are sorted based on a threshold that is typically referred to as the critical diameter, $D_{c}$. It is estimated using an empirically derived expression by Davis [14] based on a best fit model to data using hard, spherical particles:(1)$$D_{c}=1.4\cdot G\cdot N^{-0.48}$$
where G is the gap between the pillars, *N* is the period and 1/N=tanθ, where θ is the displacement angle for a device array with equal row–row distance and lateral pillar separation. The period is N=λ/Δλ, where λ is the pillar center–center distance and Δλ is the row shift.

A schematic of a DLD unit cell is seen in Figure 1A. It displays the designs of the two devices used in this work. All experiments are performed in a device (device #1) where *G* = 2.8 µm and *N =* 50, giving $D_{c}$ = 0.60 µm except where otherwise stated. All relevant array parameters for both devices are found in Appendix A.

The arrays in our devices are designed with an aspect ratio of 1:*N* (width:length) so that particles that follow the full displacement angle θ are displaced from one side of the array (at the entrance) to the opposite side of the array (at the exit). Inlet and outlet distributions are plotted normalized to the device width, with the sample entering at the zero side and being displaced towards 1.

### 2.2. Device Fabrication

The devices were designed using the layout editing software L-Edit 16.02 (Tanner Research, Monrovia, CA, USA). The devices were fabricated in polydimethylsiloxane (PDMS, Sylgard 184, Dow Corning, Midland, MI, USA) with a glass substrate using standard replica moulding [15]. The master mould was made in SU-8 2015 (MicroChem, Newton, MA, USA) using UV-lithography (Karl Süss MJB4, Munich, Germany) with a photomask from Delta Mask (Delta Mask, Enschede, The Netherlands). Directly following the UV-lithography, the mould was coated with an anti-sticking layer of 1H,1H,2H,2H-perfluorooctyltrichloro-silane (ABCR GmbH & Co. KG, Karlsruhe, Germany). PDMS structures were cast and access holes were punched, followed by air plasma (Zepto, Diener electronic GmbH & Co. KG, Ebhausen, Germany) treatment to activate PDMS and glass surfaces. Finally, the devices were bonded to glass substrates. Cover slides of 170 µm thickness (#1.5H) were used to enable imaging with a high NA oil immersion objective.

### 2.3. Sample Preparation

DNA samples, ladder DNA (0.25 kbp–10 kbp, GeneRuler 1 kbp, Thermo Fischer Scientific, Waltham, MA, USA), 5 kbp DNA sample (NoLimits, Thermo Fischer Scientific, Waltham, MA, USA), bacteriophage lambda DNA (λ DNA, 48.5 kbp, Life Technologies, Carlsbad, CA, USA) and bacteriophage T4 DNA (T4GT7, 165.6 kbp, Nippon Gene, Tokyo, Japan) were stained for 2 h at 50 °C using YOYO-1 Iodide (491 ex/509 em, Thermofisher Scientific, Waltham, MA, USA) or YOYO-3 Iodide (612 ex/631 em, Thermofisher Scientific, Waltham, MA, USA) at a 10:1 DNA base pair-to-dye molecule ratio. The stained DNA samples were stored at 4 °C for a maximum of one week before use. The DNA ladder is composed of 14 lengths (in kbp): 0.25, 0.5, 0.75, 1, 1.5, 2, 2.5, 3, 3.5, 4, 5, 6, 8 and 10. The running buffer consisted of either 1× Tris EDTA (1× = 10 mM Tris-HCl and 1 mM EDTA, pH 8), or 5× Tris EDTA and 3% betamercaptoethanol (BME), or Milli-Q^®^ water. To prevent non-specific sticking of the DNA or dye molecules to the channel walls, Pluronic^®^ F-127 (MW~12,500 Da) was added to both the running buffers and the samples to a concentration of 10 μg/mL or 0.001% (*w/v*, final concentration). All solutions except the DNA samples and the dye solutions were filtered through 0.2 µm pore filter before mixing. The λ DNA samples were kept at 65 °C for 10 min to remove concatemers followed by rapid cooling in an ice bath. A complete list of the samples that we used in our experiments, their buffers and some of their physical properties can be found in Appendix A.

In the experiments, two ionic strengths of the buffer were used (6.1 mM and 43.6 mM, 1× Tris-EDTA (TE) and 5× TE and 3% BME, respectively). At first, we used the higher ionic strength to improve homogeneity of the fluorescent staining (as showed by Nyberg et al. [16]). Note that the ionic strengths of the samples differed slightly from the buffers in the sheath flow. However, as the sheath flow and sample flows were mixed, these differences are decreased. We used salt at dramatically different concentrations to increase the difference of the two T4 DNA samples in terms of *C*/*C** in order to explore the effect of extreme values of *C*/*C**, see Figures 3C and 4. Due to the swelling of the polymer at low salt, *C*/*C** is increased for a given DNA concentration.

### 2.4. Fluidics

Flow was generated in the device using nitrogen at different overpressures applied at the inlets with an MFCS-4C pressure controller (Fluigent, Paris, France) for pressures up to 1 bar and a custom-built manifold for pressures between 1 bar and 3 bar. Flow was measured using a flow sensor (Flow rate platform with flow unit S, Fluigent, Paris, France) that was connected to the outlet reservoir. The outlet reservoirs were kept at ambient pressure. The devices were cleaned by rinsing with the running buffer for 10 min each time prior to running the sample. The measured ambient temperature was 21.8 ± 0.3 °C.

### 2.5. Microscopy

All images were acquired using an inverted Nikon Eclipse Ti microscope (model TI-DH Nikon Corporation, Tokyo, Japan) with an electron-multiplying (EMCCD) camera (iXon 897-DU Andor Technology, Belfast, Northern Ireland) and SOLA light engineTM (6-LCR-SB, Lumencor Inc, Beaverton, OR, USA) with FITC or Cy5 filter cubes. Objectives 2× (Nikon Plan UW, NA 0.06, Field of View (FoV) of 4096 µm), 10× (Nikon Plan Apo λ, NA 0.45, FoV of 819 μm) and 100× (Nikon Plan Apo VC, Oil Immersion, NA 1.4, FoV of 82 μm) were used and videos were captured at 10 to 201 frames per second. For dual-colour imaging and polarization microscopy, an Optosplit (Cairn Research Ltd., Kent, UK) was used with the appropriate filter sets.

The plotted lateral distributions of the DNA are based on integrating the fluorescence intensity at the inlet and outlet regions. They have been normalized so that the total area under each curve is the same for each flow velocity for each experiment. In this way they can be interpreted as probability distributions. The full lateral range (0 to 1) corresponds to the full width of the microfluidic channel. See Appendix A for more details on the image processing.

## 3. Results

We demonstrate high throughput of long DNA displacement and separation in micro-scaled DLD devices at flow velocities that are extremely high relative to those previously reported (tens of mm/s compared to tens of µm/s, see the full comparison together with details of differing flow velocity measurement approaches in Appendix A). Our method works well for dilute sample concentrations and can be enhanced at higher concentrations. We speculate that the interaction of individual DNA molecules plays an important role in the concentration dependence of the performance of our devices, see discussion section. Therefore, we specify all DNA concentrations relative to the overlap concentration, *C/C** [17]. *C** depends on the contour length, *L*, as *C**
$\propto L^{\left(1-3\nu \right)}\approx L^{-0.8}$, where *ν* is the Flory exponent, *ν* = 0.5877 [18]. The presence of polymers in a solution can have a large effect on rheological properties, indeed typically also for DNA. The relative importance of viscoelastic behavior under various flows of polymer solutions is often quantified by the Weissenberg or Deborah numbers. Here, the Deborah number is selected over the Weissenberg number since it better describes elastic responses to transient deformations, such as those experienced by the DNA molecules as they flow through a pillar array [19]. We give estimates of the Deborah number ($De=\tau_{Zimm}/\tau_{flow}=\tau_{Zimm}\cdot u/L)$, where L is the array pitch, u is the mean flow velocity, $\tau_{flow}=L/u$ is the observation time and $\tau_{Zimm}$ is the Zimm relaxation time for a given length of the DNA [20] for the results presented below. We give the Deborah number for all our results except those involving very short DNA where the Zimm relaxation time is not applicable.

We present our results as follows. First, we demonstrate successful separation of two long DNA species as well as separation of DNA mixtures of relevant molecular sizes. Secondly, we show how the DNA of different lengths and concentrations is displaced as a function of flow velocities in the DLD array. Finally, we investigate important details of the separation to better understand the performance and the basis for the separation.

### 3.1. Isolating Two Long DNA Populations from Each Other

We demonstrate high-throughput separation using a mixture of T4 phage (166 kbp) and λ phage (48.5 kbp) DNA samples, see Figure 1B. At the highest tested pressure, 3 bar, the mean flow velocity, u, is approximately 34 mm/s and the sample throughput approximately 760 ng/h. u is calculated using the measured flow rate, *Q*, and the cross-sectional area at the narrowest gaps in the arrays, *A*, as u=Q/A. Based on the outlet distributions given in Figure 1C, it is possible to select the position of the outlet channels carefully in order to optimize any collected samples based on the requirements for size, purity and recovery that are necessary for each particular application, see Appendix A. For example, for two equally wide outlets where u = 34 mm/s, the purity rate of the 166 kbp sample in a rightward outlet would be 97% and the recovery rate 70%. Based on the observed outlet distributions, it is clear that close to 100% purity can be attained if the cut-off is moved towards greater displacement. Of course, this is at the cost of a decrease in recovery rate. We achieve this degree of separation despite instabilities observed in the sample stream, see Appendix A.

### 3.2. Isolating Long DNA from a Background of Short DNA

To mimic a situation withg a DNA of length of interest against a background of short DNA, we demonstrate the separation of λ phage DNA (48.5 kbp) at different concentrations from a DNA ladder (0.25 kbp to 10 kbp) at a fixed relatively high concentration.

The shorter DNA sample represents a high concentration (100 µg/mL) of unwanted DNA fragments while the long DNA represents long-read samples to be purified. Note that the concentration of the short fragments is well below the overlap concentration ($C_{10\;\mathrm{kbp}}^{*}\;$≈ 140 µg/mL, calculated for the ionic strength of the sample after mixing with the sheath flow (1× TE)) of the longest fragment length (10 kbp) so that we expect negligible overlap effects from the high concentration of the shorter fragments. We show the effect of the long DNA concentration on the separation in Figure 2. We have employed a dual-colour imaging setup and stained the two samples with different dyes for simultaneous imaging of the two samples. For the dilute sample (*C* ≈ 3.7 µg/mL or *C/C** ≈ 0.07), a high degree of separation is achieved up to approximately 1.7 mm/s and partial separation at approximately 4.2 mm/s, see Figure 2A.

For higher concentrations of the target λ DNA, both samples displayed similar trajectories and no separation could be achieved, see Figure 2B,C. Interestingly, while no displacement occurred at higher flow velocity for the dilute sample, displacement was achieved at all flow velocities tested for the concentrated samples, even at the highest flow velocity.

We observe that the high concentration of λ DNA influences the distribution of the short DNA which may be a possible effect of hydrodynamic molecule–molecule interactions or entanglement as discussed in Section 4 below.

### 3.3. Displacement of DNA as a Function of Length and Concentration

To investigate the potential of separation and increase in concentration, we observe the displacement behaviour of individual DNA specimens of three lengths (5 kbp, 48.5 kbp and 166 kbp) at lower and higher concentrations as a function of flow velocity.

A general observation is that the short, 5 kbp DNA is not at all displaced, see Figure 3A, irrespective of concentration and flow velocity. However, higher flow velocities decrease any effect of diffusion.

For the longer DNA, 48.5 kbp and 166 kbp, larger concentrations make the displacement more robust to increases in flow velocity. As the flow velocity is increased, the displacement does not decrease as much for higher concentrations as for lower concentrations. However, higher concentration for the intermediate-sized DNA, 48.5 kbp, decreases the initial displacement even at low flow velocities, see Figure 3B, but then this displacement exhibits a lesser relative change compared to the low-concentration case.

In Figure 3C, the separate behaviours of the three independently run samples are shown together. They have different trajectories for a mean flow velocity, as high as at least 2.6 mm/s, which corresponds to a throughput of approximately 2 µL/h or 75 ng/h.

### 3.4. Limitations of Flow Velocities and Concentrations

While Figure 3 shows the average outlet distributions, Figure 4 illustrates the detailed behaviour of the flow of the DNA samples in one experiment for the long DNA, T4 phage DNA (166 kbp), at the higher concentration (23 µg/mL, *C*/*C** >> 1, in a 0.26× TE buffer) compared to the lower concentration (2.3 µg/mL, *C*/*C** << 1, in a 4.9× TE buffer). Here, the different buffers are used to enhance the difference in *C/C**. Two important observations can be made regarding the behaviour of the higher concentration. Firstly, the overall displacement is larger across the range of flow velocities tested (Figure 4A bottom row) and the sample stream is more focused. Secondly, at sufficiently high flow velocities O(1)  mm/s), we observe instabilities in the stream of the displaced DNA. These instabilities take the form of variations in the lateral position of the sample stream over time, see Figure 4A, and in the concentration of the DNA along the length of the displaced sample stream, see Figure 4B,C and Appendix A. Note that, despite these variations, we can still obtain a significant displacement of the sample that can be leveraged for separation.

### 3.5. Influence of Periodicity

To test the importance of periodicity, *N*, we investigated the performance of long DNA (T4 DNA, 166 kbp) displacement with a device developed for previous work with *N* = 20 (device #2), instead of *N* = 50 (device #1), with which the main experiments have been conducted. We selected a device with a gap size so that the critical diameter was kept approximately the same (0.60 µm and 0.74 µm for device #1 and #2, respectively, calculated using Davis’ equation, see Equation 1). We compared the critical diameters to the estimated radius of gyration of T4 DNA, ~1.4 µm and the corresponding diameter of 2.8 µm. The concentration was kept low (2.3 µg/mL) to minimize any concentration-dependent effects that could have been of different magnitude in the two devices. See Figure 5 for the lateral outlet distributions with the two devices. While the long 166 kbp sample could be fully displaced at flow velocities up to 160 µm/s, and partially displaced at speeds up to 4.1 mm/s in device #1, there was not even partial displacement at 170 µm/s for device #2.

### 3.6. Dynamics of the Conformation of the DNA

A simplified understanding of DNA separation in DLD is that the DNA maintains an approximate spherical conformation with a radius corresponding to the radius of gyration. Based on this, it is natural to expect any stretching of the DNA to be detrimental to the sorting. When imaging individual DNA molecules moving through the array while being displaced, we can observe that the molecules do not assume any spherical shapes; instead, they exhibit rich conformational dynamics, see Figure 6. Furthermore, they are stretched out at higher applied pressures, see Figure 7, under which we can clearly observe that they are displaced, based on Figure 3C and Figure 4A, as well as the fact that the Region of Interest (ROI) is placed at the particular outlet part which can be reached only by displacing DNA.

## 4. Discussion

We can displace long DNA at unprecedented flow velocities and throughputs. We believe the main factor that lets us achieve this is a smaller displacement angle, i.e., a greater *N* in the pillar array, see Figure 5, and larger pillars compared to previous work. We use a longer device providing longer residence times and more contact for lift forces and rotational effects. 

The throughput is ultimately limited by viscoelastic effects occurring at high concentrations and flow velocities. Hydrodynamic molecule–molecule interactions and entanglement may influence the group behaviour of molecules at high concentration, leading to both increased and decreased displacement under different, specific conditions. We explore the limits in concentration and flow velocity, and show that these effects may be deleterious for separation.

### 4.1. High Throughput

We demonstrate a working separation throughput up to 24 µL/h or 760 ng/h of λ DNA and T4 DNA. The effective volumetric throughput is approximately two orders of magnitude higher than previous DLD separation work of long DNA, whereas the sample throughput is approximately three orders of magnitude higher. While the main aim of our work is to increase the throughput, we can also compare the time it takes from loading until the first separation takes place, approximately 0.7 s for λ-DNA and T4 DNA at 3 bar (residence time of the array at 34 mm/s in a 22.9 mm long array), compared to the 15 s stated for previous work with on-chip pulsed-field electrophoresis [8]. See the full comparison in Appendix A. The high throughput makes it possible to collect the separated species in a high enough concentration for genetic analysis in a very short time. Others have chosen a different approach towards high throughput by massively parallelizing nanoDLD devices (up to 31,160 arrays in parallel [21]). While this method seems to work well, such parallelization is highly complex, expensive and limited to labs with advanced fabrication facilities. Our method, on the other hand, requires only a single device which is fabricated with standard soft lithography.

### 4.2. High Concentration

At concentrations above the overlap concentration (*C/C** >> 1), the displacement of T4 molecules is seen to increase for low flow velocities (Figure 3C and FigureFigure 4A). At higher flow velocities, instabilities are seen as variations in the degree of displacement over time (millisecond time scale) and in the local concentration of the DNA along the sample stream (length scale 0.5 mm), see Figure 4. Based on our low Reynolds numbers (upper limit estimated at *Re*~0.1 using density and viscosity for water, together with the centre-to-centre distance of the pillars), we believe that these observed effects have their origins in entanglement and viscoelastic phenomena that become influential at high concentrations. What is more, hydrodynamic molecule–molecule interactions play an increasing role at higher concentrations and, while we are below the average sample concentrations at which topological entanglement becomes prevalent, local increases in the concentration, on the scale of the pillars and gaps, together with non-adiabatic deformations of single and groups of molecules, may well lead to entanglement.

Entangled or strongly hydrodynamically interacting molecules could act together as a “super-molecule” with a larger effective group size than individual molecules. This mechanism would be dynamic, with the super-molecules existing only transiently, but still, on average, leading to increased separation. These effects could contribute to the broadening of the short DNA fragment streams seen in Figure 2B,C as the concentration of λ DNA is increased. Here, the longer λ DNA fragments might be carrying the shorter fragments with them as they displace.

Well-defined flows are essential for the functioning of DLD separations. Deviations from laminar flow, such as those due to vortex formation in flows with a significant inertial component, have been shown to negatively affect separations. Dincau et al. showed a worsened separation efficiency at moderate *Re* (4 < *Re* < 34) for angled airfoil-shaped pillar arrays. Interestingly, with a neutral-angled airfoil pillar which avoided vortex formation, the particle trajectories were only shifted and not broadened [22]. In flows of polymers, the build-up of elastic forces, as molecules follows highly curved streamlines, has been shown to lead to spatial and temporal flow fluctuations even at low *Re* [23]. It is likely that such mechanisms are at play here in our high-concentration separations and they place constraints on the maximum concentrations and flow velocities that can be used depending on the device, sample and the application. Viscoelastic effects that might influence displacement are discussed below.

### 4.3. Displacement Mechanism

In the following, we discuss mechanisms that we identify as potential contributors to the lateral displacement of the DNA.

Polymers are known to elongate from a coiled conformation as a response to a shear or elongational flow. In pillar arrays, such as in DLD devices, the flow is a combination of both shear and elongational flow. High shear and elongational rates have been reported to reduce the effective size of the polymers in DLD, which causes them to follow a zig-zag trajectory rather than a displacement trajectory. Chen et al. found that when polymers elongate to a sufficient degree so that their short axis is shorter than the critical size, they will follow a zigzag trajectory and stop displacing [13]. They were unable to displace T4 DNA in their DLD devices unless they used a carefully tuned concentration of polyethylene glycol (PEG) to gently compact the DNA. Note that, with the compaction, they were only able to displace the DNA molecules up to 40 µm/s (compared to our flow velocity of up to 34 mm/s for T4 DNA displacement). Wunsch et al. found, contrary to our results, that, when the pillar gap reaches the micrometre scale, DNA displacement is not observed even with a relatively low flow velocity [11]. In contrast, we show that, despite considerable elongation, it is possible to displace long DNA molecules in devices with micrometre-scale gaps. 

We can estimate the flow velocity $u_{transition}$ at which the *De ≈ 1* and where molecules are expected to elongate [24] as $u_{transition}\approx De\cdot L/\tau_{Zimm}$. For the molecules that are displaced (48.5 kbp and 166 kbp), we have with *L* = 10 µm and $\tau_{Zimm}$ ≈ 1 s, $u_{transition}$ ≈ 10 µm/s. All of our experiments are performed at u > $\;u_{transition}$ and we can, therefore, expect elongation to be present at various degrees for all measured pressures.

We ascribe our remarkable separation performance to several factors. The larger gaps between pillars in our device (much larger than the persistence length of the DNA, 50 nm to 100 nm, depending on the salt) allow for a greater range of dynamic molecular conformations compared to devices with nanoscale gaps. As described by Smith and Chu [24], when a molecule is stretched suddenly in an extensional flow at a rate that exceeds its relaxation rate, it can adopt a range of nonequilibrium conformations. In our case, not only the large *De* but also the complexity of the flow field through the pillar array, which is far from simple extensional flow, lead to a rich variety of molecular conformations. Because the molecules are long in comparison to the pillar size and pitch, they can span multiple pillars and gaps, and experience forces along their lengths that vary greatly in degree and direction. The long DNA molecules are able to form multiple blobs in the areas of low flow velocity between rows of pillars that are connected with stretched segments of the molecule (Figure 6A and Figure 7 top panel). The molecules also become stretched perpendicular to the flow direction (Figure 6B and Figure 7 middle panel). Long molecules have space to rotate (Figure 6C). DNA molecules are known to continuously rotate in a shear flow [25]. Indeed, such rotation is visible in Figure 6 and in Appendix A. The rotation will most likely increase the likelihood of displacement as it has been demonstrated for cell clusters [26], as well as what we previously observed for bacteria [27] and parasites [28] that were sorted based on their shape. While it is difficult to ascertain the exact contributions of these effects to the enhanced displacement of the molecules, they show that the molecules are being influenced by the entire flow field between the pillars and even across multiple pillars/gaps simultaneously. This indicates that a description of the displacement mechanism of long polymers in micrometre-scaled DLD devices cannot be fully captured via the assumption that the molecule behaves like a prolate ellipsoid with a short axis that monotonically decreases with increased shear rates.

Possibly based on the role played by the rich dynamics of the DNA as it moves between the pillars, we find that the effect of *N* might not be well described for the DNA by the Davis equation, see Equation (1). The Davis equation has been empirically derived and describes clearly the effect of gap size and displacement angle (*θ* = tan^−1^ (1/*N*)) on the critical size of a DLD array for hard spheres. The effects of the gap size, *G*, on the sorting of the DNA in DLD has been explored previously [11]. To investigate the effect of the periodicity, *N*, we performed a preliminary test by comparing two devices with similar $D_{c}$ but with very different *N*. The device used for the bulk of the work presented here has *G* = 2.8 µm, *θ* = 1.1°, *N* = 50 and an approximate $D_{c}$ = 0.60 μm ($D_{c}$ similar to those employed in previous work (~0.7 μm for [10])). We were able to compare the displacement of the DNA in this device to that in a device designed by us for other work where *G* = 2.24 µm, *θ* = 2.9°, *N* = 20 and an approximate $D_{c}$ = 0.74 μm. While we might expect a small difference in the displacement of molecules at varied flow velocities due to the small difference in $D_{c}$, we actually find that the device with *N* = 20 is only able to displace T4 DNA on the order of tens of micrometres per second while the device with *N* = 50 can displace T4 at millimetres per second as shown in Figure 5. The large disparity between these results is surprising. Since the diameter of the molecules (twice the radius of gyration) is 2.8 µm and the critical diameters in both devices are, while slightly different from one another, much smaller than this, one would expect them to perform more similarly. Modifications to the Davis equation that help to predict the behaviour of the DNA and other non-spherical, non-rigid particles would be very useful during the designing of future devices. While further studies are required, the preliminary results presented here indicate that *N* will be an important parameter to explore. What is more, for a given $D_{c}$*_,_* larger *N* allows for larger *G*, which means that devices are less prone to clogging,

There are also other effects that may contribute to a length-dependent displacement. We also cannot exclude the possibility that various lift forces (entropic, hydrodynamic and elastic) also play a role. Close to the channel wall, the number of possible conformations is lower. Thus, entropic effects lead to migration away from the wall on the scale of the radius of gyration. The hydrodynamic wall lift force, $F_{w}$, will push the DNA away from the wall due to its flexibility [29]. The elastic lift force also pushes the DNA away from the wall, as has been shown for the DNA in microchannels [30] and for particles in viscoelastic media in DLD arrays [31]. The elastic lift force is predicted to increase with the polymer concentration and flow velocity [32].

Some effects act to reduce the displacement. In curved streamlines, the hoop stress causes the polymers to move in the radial direction [33]. Diffusion of the particles has also been shown to reduce the displacement effect [34], but we expect that to be a problem only for short molecules and for low flow velocity, which is not very relevant for the experimental conditions that we have explored.

### 4.4. DNA Fragmentation by Hydrodynamic Shearing

Flow across a constriction at high flow rate has been shown to result in chain scission of long DNA [35,36,37,38]. To test whether there is any significant hydrodynamically induced fragmentation, we collected the resulting sub fractions from a device at the highest run pressure and ran the samples again. While the original sample contained a considerable portion of shorter fragments, after collecting the deflected sample and running it a second time, no shorter fragments were observed. The results can be found in Appendix A.

While we did not observe significant fragmentations, even at the highest run pressures, it is possible that longer molecules or higher pressures could lead to fragmentation. The constriction shape [35] and length [37] have been shown to affect the fragmentation. A gradual increase in the extensional rate and small constriction lengths, as occurring in our devices, has been shown to reduce the fragmentation significantly. While a certain sample loss is manageable, significant hydrodynamic fragmentation sets an upper limit on flow velocity and, consequently, that of the throughput. Finally, photodamage of the DNA must be taken into account for those cases where the separation is monitored using fluorescence microscopy.

Using larger gaps has been shown to reduce the hydrodynamic fragmentation of long DNA [36]. By using micrometre-scaled gaps instead of hundreds of nanometres (e.g., used in nanoDLD [11]) we reduce the shear rate and, consequently, increase the throughput by approximately 100-fold. Using microscale devices also reduces the chances of clogging, which can otherwise be a considerable problem when working with complex samples such as lysate.

## 5. Conclusions

We are able to achieve lateral displacement of DNA at an unprecedented flow velocity of 34 mm/s and throughput of 760 ng/h. In contrast to the prevailing view in the literature, we show that at such high flow velocity, the DNA molecules are extended and, thus, the separation is not dependent on the radius of the coiled conformation.

We accomplish throughputs high enough to produce large enough samples for standard analysis techniques in a short time period. This applies to applications involving the separation of long DNA from short DNA, as well as applications involving increasing the concentration of existing DNA samples. We fabricate the devices using standard soft lithography without the need for complicated nanofabrication or parallelization. The high throughput, together with the simplicity, makes the technique practical and accessible in both industrial and academic settings.

We show that separations can be improved by increasing the concentration but that this can ultimately also lead to various elastic flow instabilities under some conditions that can be deleterious to the separation. However, further investigations are needed to elucidate the detailed dependence of the flow properties, the sorting capabilities as a function of the concentrations and the compositions of the DNA solutions used. Furthermore, it would be interesting to increase the applied pressures to beyond the 3 bar that our setup was capable of delivering to probe the ultimate limit of the sorting.

Devices with larger array periodicity *N* (for a given $D_{c}$) are better at displacing long molecules at high flow velocities. For a given $D_{c}$, further increasing *N* means larger gaps, reduced clogging, lowered fluidic resistance and potentially further increases in throughput. However, the reduced angles mean longer devices are needed to achieve the same spatial separation. Future work will explore further improvement in device functionality via the optimisation of *N*. In this way, ultimate limits can be identified and optimal conditions can be selected for each specific application (separation of long molecules with varied length at high concentration and separation of rare long molecules from a large background of short fragments).

## Figures and Tables

**Figure 1 micromachines-13-01754-f001:**
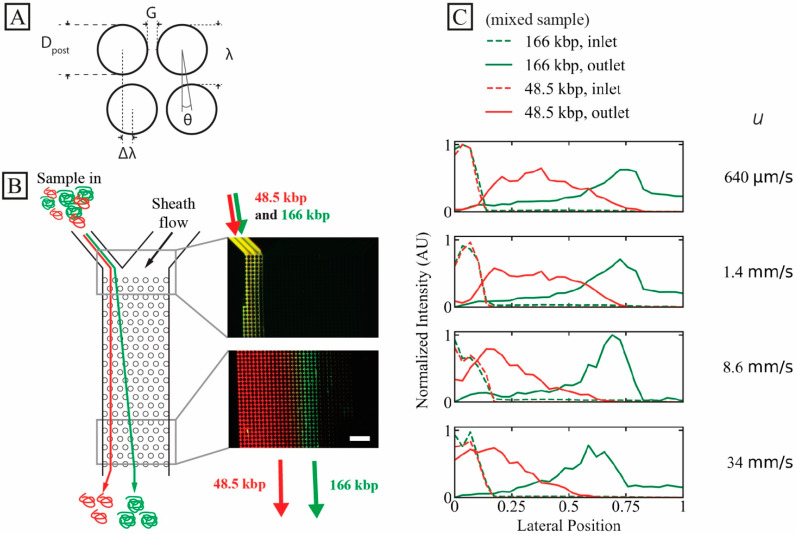
High-throughput long DNA separation in a microscale DLD array at ultra-high flow velocity. (**A**) Schematic of DLD unit cell with relevant array parameters. (**B**) Separation of 166 kbp (green, 8 µg/mL, *C/C** ≈ 0.53) and 48.5 kbp (red, 24 µg/mL, *C/C** ≈ 0.62). Time-averaged (1 min) micrographs show the inlet (top) and outlet (bottom) of the array at the highest run pressure drop (3 bar, *u* ≈ 34 mm/s). The measured sample inlet flow rate at this pressure was 24 µL/h, corresponding to a sample throughput of 760 ng/h. The samples have been stained with YOYO-1 (green) and YOYO-3 (red). (**C**) Lateral inlet and outlet distributions at different flow velocities (corresponding to a range of Deborah numbers $1.1\times 10^{2}$ to $5.7\times 10^{3}$ for 48.5 kbp DNA and $9.5\times 10^{2}$ to $5.0\times 10^{4}$ for 166 kbp DNA). The curves have been normalized so that the areas of the distributions (inlets or outlets) are the same. The running buffer was 0.9× TE. Note that the presented overlap concentrations, *C**, are calculated for the samples independently and not based on a combined effect of the two samples. The scale bar is 100 µm. A video showing the raw data for the highest applied pressure (3 bar), for the two DNA components, can be found in the Appendix A.

**Figure 2 micromachines-13-01754-f002:**
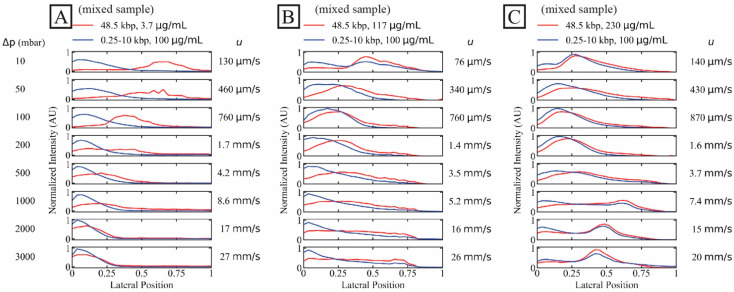
Separation of long DNA (48.5 kbp) from short DNA (0.25 kbp to 10 kbp) as a function of three long DNA concentrations. The short DNA concentration was set constant at 100 µg/mL while the long DNA concentrations were (**A**–**C**) 3.7 µg/mL (dilute, *C*/*C** ≈ 0.070), 117 µg/mL and 230 µg/mL (*C*/*C** of approximately 2.6 and 4.1, respectively, both semidilute). Images were collected simultaneously in two colours. The curves have been normalized so that the areas of the distributions are the same across each panel. Note that the salt concentrations differ slightly between the three samples. However, these differences become insignificant as the buffer in the sheath flow is of the same salt concentration between the runs, see Appendix A. A video showing the raw data for the lowest and the highest applied pressures, for the two DNA components and for the three different concentrations of the λ phage DNA can be found in the Appendix A.

**Figure 3 micromachines-13-01754-f003:**
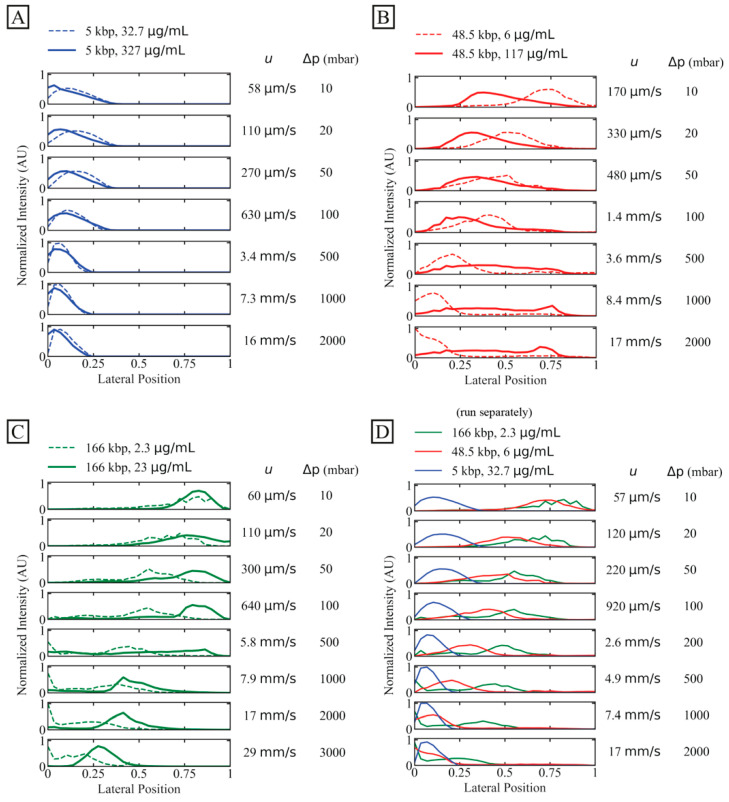
Displacement of 166 kbp, 48.5 kbp and 5 kbp DNA. We characterize outlet distributions comparing low and high concentrations of differently long DNA. The presented data represent averages of several experiments. (**A**) 5 kbp distributions with two dilute samples (32.7 µg/mL or *C/C** ≈ 0.083, dashed line, and 327 µg/mL or *C/C** ≈ 0.64, solid line). (**B**) 48.5 kbp at dilute (6 µg/mL or *C/C** ≈ 0.065, dashed line) and semidilute (117 µg/mL or *C/C** ≈ 1.4, solid line) concentrations. Note that an additional peak emerges for the semidilute sample at 8.4 mm/s and 17 mm/s. (**C**) 166 kbp with two dilute samples (2.3 µg/mL or *C/C** ≈ 0.064, dashed line, and 23 µg/mL or *C/C** ≈ 8.8, solid line). (**D**) Dilute samples from panel (**A**–**C**) plotted together. The stated flow velocities are the average mean flow velocities between the two samples in the same plots. The flow velocities correspond to Deborah numbers in the range 9.6 to $5.6\times 10^{4}$. The curves have been normalized so that the areas of the distributions are the same across each panel. Note that the salt concentrations differ slightly between the samples. However, these differences become insignificant as the buffer in the sheath flow is of the same salt concentration between the runs (except the high concentration 166 kbp sample where water was introduced to the buffer inlet), see Appendix A.

**Figure 4 micromachines-13-01754-f004:**
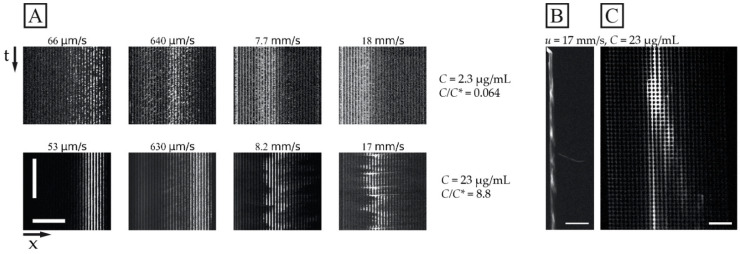
Displacement of 166 kbp long DNA corresponding to the data shown in Figure 3C. (**A**) Kymographs at the array outlet for dilute and semi-dilute samples of 166 kbp long DNA (2.3 µg/mL or *C*/*C** ≈ 0.064 in 4.9× TE buffer, and 23 µg/mL or *C*/*C** ≈ 8.8 in 0.26× TE buffer) for four pressure values (10 mbar, 100 mbar, 1 bar and 2 bar, from left to right). The sample streams are stable for the dilute sample and at 53 µm/s and 630 µm/s for the semidilute sample. However, they are highly unstable and undulating at 8.2 mm/s and 17 mm/s. The brightness and contrast settings have been enhanced independently for each image. The exposure times are 100 ms. The horizontal and vertical scale bars are 200 µm and 15 s, respectively. (**B**,**C**) displays fluorescence micrographs of the array for the 23 µg/mL sample at *u* = 17 mm/s (*De* = $2.2\times 10^{4}$). (**B**) shows a pulsing sample stream at the array inlet whereas (**C**) shows a single wave peak close to the array outlet. The video corresponding to panel (**B**) can be found in the Appendix A. Scale bars are 500 µm and 100 µm for panels (**B**,**C**), respectively.

**Figure 5 micromachines-13-01754-f005:**
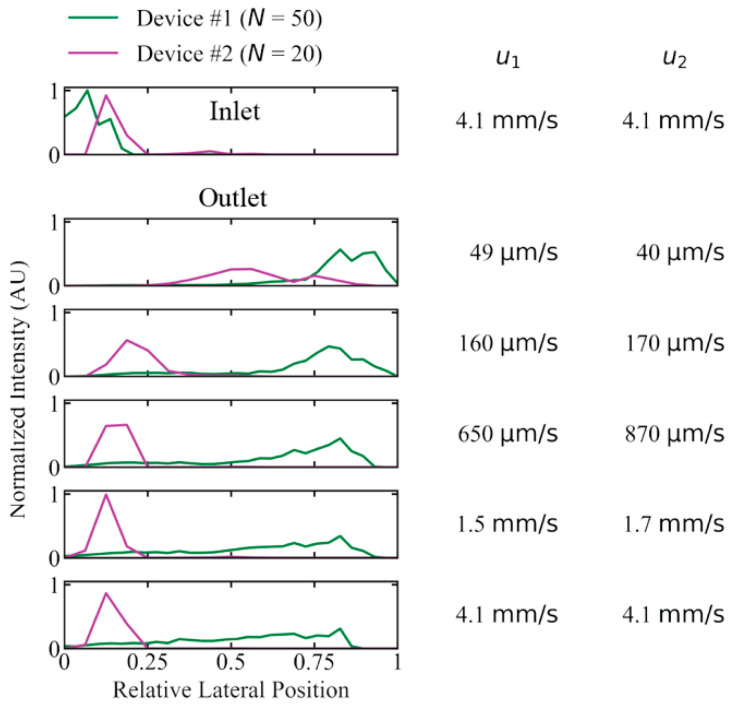
Difference in displacement performance due to the periodicity, *N*, of the pillar array for device #1 (*N* = 50) and #2 (*N* = 20) for 166 kbp long DNA (2.3 µg/mL or *C*/*C** ≈ 0.15) in a running buffer of 0.9x TE. Device #1 exhibits a much larger degree of displacement at high mean flow velocity compared to device #2 (*u_1_* and *u_2_* are given for device #1 and #2, respectively). Deborah numbers range from $7.1\times 10^{1}$ to $5.9\times 10^{3}$ in device #1 and from $8.6\times 10^{1}$ to $8.8\times 10^{3}$ in device #2. The curves have been normalized so that the areas of the distributions are the same for each device. Because the two arrays have different widths, the lateral positions are relative and not absolute as for the other figures. See Appendix A for more details on the device designs.

**Figure 6 micromachines-13-01754-f006:**
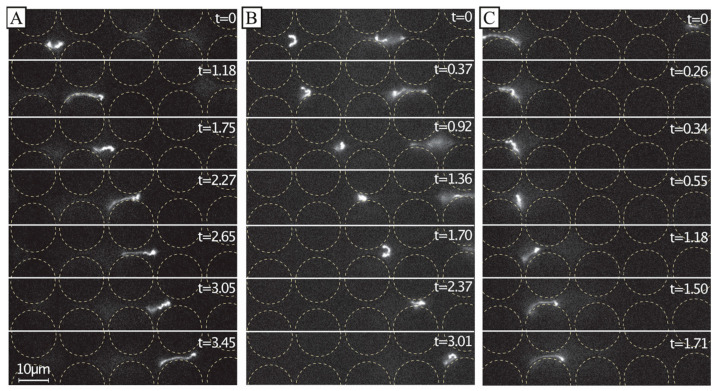
Dynamics of dilute 166 kbp long DNA strands (*C =* 0.01 µg/mL, 5× TE buffer). We show motion of three individual molecules at the end of the device and laterally displaced (**A**–**C**) *u* ≈ 21 µm/s (*De* = $2.7\times 10^{1}$). The images show the evolution of the shape of the molecules at different time scales. The frames are sequential but with different temporal gaps. The exposure time was 19.3 ms. The videographs corresponding to the panels can be found in the Appendix A, see Video S4. Flow is left to right. Time stamp unit is seconds. Scale bar is 10 µm.

**Figure 7 micromachines-13-01754-f007:**
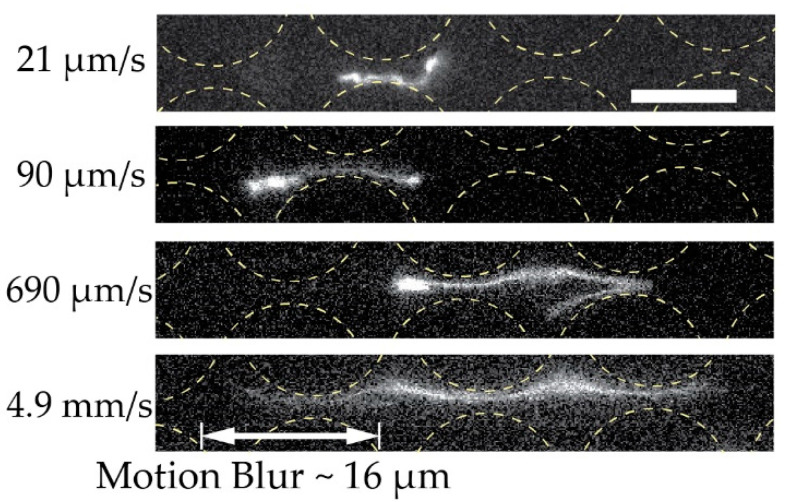
Extension of dilute 166 kbp long DNA (*C =* 0.01 µg/mL, 5× TE buffer). We show the extension of four individual molecules at four different mean flow velocities (*De* of $2.7\times 10^{1}$, $1.2\times 10^{2}$, $8.9\times 10^{2}$ and $6.3\times 10^{3}$) at the end of the device and laterally displaced. The exposure times were 19.3, 3.00, 4.36 and 3.17 ms for the micrographs top to bottom. Flow is left to right. Scale bar is 10 µm.

## Data Availability

Data available in a publicly accessible repository. The data presented in this study are openly available in Harvard Dataverse at https://doi.org/10.7910/DVN/SURRJQ (accessed on 12 October 2022).

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
