# Peer review of "High-Throughput Separation of Long DNA in Deterministic Lateral Displacement Arrays"

_micromachines, 2022, doi:10.3390/mi13101754_

Round 1

Reviewer 1 Report

The manuscript describes separations of DNA according to size using a deterministic lateral displacement (DLD) device.  The device itself is a series of posts/obstacles through which a solution of DNA is pumped.  The DNA is introduced on one side of the device and then spreads in the lateral direction as it is transported in the flow direction.  Larger DNA molecules move further in the lateral direction than do smaller ones, consequently a size separation can be made by segregating the streams at the outlet.

Size separation using DLD depends on the net lateral displacement of particle upon collision with a post.  The net displacement for each collision is larger for big versus small particles.  Consequently, small particles continue relatively directly down the channel length while the bigger particles will translate a relatively large distance in the lateral direction by the time they reach the outlet.

The big question is why the method works for DNA under the conditions described in this manuscript.  The largest DNA are shown to stretch a significant amount in response to the large flow rates used in the experiments (see figures 6 and  in the manuscript), which should reduce the effectiveness of the separation.  Indeed, flows that are too high do seem to reduce the effectiveness, as shown in figure 2A.

In section 4.3, the authors speculate on the reasons for the (unexpected) effectiveness of the separation.  However, the discussion is difficult to follow as the quantities $N$, $G$, $D_c$, and $\theta$ are never defined that I can see in the document.  Still the authors appear to attribute the separation effectiveness to various lift forces (elastic, hydrodynamic), rather than irreversible collisions.  These forces are presumably stronger for larger DNA molecules, though in general these forces are quite weak.

I recommend that the authors address the following comments prior to publication:

* The authors discuss two devices.  We are told how the devices were made in section 2.1, but the dimensions and specifications of the devices are not given within the paper nor the supplementary materials that I can see.  This included the definitions of $N$, $G$, $D_c$, and $\theta$.  Best would be a schematic labeling the various parameters and then listing the values for each parameter for the two devices.   This information must be provided.

*  The best separation result seems to be at low concentration and low velocity.  The reduction in effectiveness with concentration is relatively easy to understand (entanglement).  However, the lift forces should generally operate better at larger velocities (really shear rates).  To better evaluate the reasons for the reduced effectiveness at higher flow rates, it would be helpful if the authors could quantify their flows in terms of a Weissenberg number and other dimensionless groups relevant to their system, such as the Reynolds number.  The Weissenberg number compares the rate of shear and relaxation times for a polymer, and the polymer is expected to uncoil at large Weissenberg numbers.  The hydrodynamic lift mechanism depends on the stretch of the DNA, for example.

* Line 98: ``0.25 kbp – 10 bp'' should probably be ``0.25 kbp – 10 kbp''.  The document overall is well-written and I found only this one typographical error.

Reviewer 2 Report

The authors propose that the separation of long DNA molecules in DLD arrays does not depend on the radius of their helical morphology, but this conclusion contradicts the deterministic lateral displacement sorting mechanism, and the conclusion lacks rigor, and the following recommendations are made:

1166-167, Define the flow rate as the ratio of the array inlet flow to the narrowest microcolumn gap, which is, however, the definition of flow velocity by Huang, Chen, and Wunsch et al[1]. did not specify the flow velocity at the narrowest gap. Will this definition have a greater impact on the comparison between flow rates?

[1] Huang L R, Cox E C, Austin R H, et al. Continuous particle separation through deterministic lateral displacement[J]. Science, 2004, 304(5673): 987-990.

[2] Chen Y, Abrams E S, Boles T C, et al. Concentrating genomic length DNA in a microfabricated array[J]. Physical review letters, 2015, 114(19): 198303.

[3] Wunsch B H, Kim S C, Gifford S M, et al. Gel-on-a-chip: Continuous, velocity-dependent DNA separation using nanoscale lateral displacement[J]. Lab on a Chip, 2019, 19(9): 1567-1578.

2287-292, In this paper, two arrays with similar critical dimensions but different periods (lateral displacement) are designed.

(1) When the critical dimension remains the same, increasing the period leads to a decrease in the lateral displacement, is it more likely to cause clogging in the microchip when the DNA concentration is higher?

(2) As a flexible particle, the deformation of the DNA molecule is affected by the pressure, viscous force, and shear force in the micro-column gap. Although the critical dimensions of the two devices are similar, their length and width are quite different. Under the condition of fully developed fluid flow, DNA Is there a big difference in the force on the molecules? Please use the Reynolds number or other dimensionless numbers for further explanation.

3312-317, Aiming at the deformation movement of DNA molecules in the gaps of the micropillars, the authors propose that its stretching is beneficial to improve the sorting efficiency. Combined with Fig. 6, Fig. 7, and the attached video 4, when the DNA molecules are laterally displaced from the upper column of micro-columns to the following micro-columns with the fluid flow direction, their shape changes from strip-like to particle-like, according to the rotation radius of T4-DNA (1.1 μm-1.7μm) is larger than the critical radius of the array (0.6μm) and the lateral displacement still occurs. It can be speculated that the rotation radius of the DNA molecule plays a fundamental role in its lateral displacement. Therefore, I think the conclusions proposed by the author lack rigor.

4A higher flow rate is likely to lead to the breakage of DNA molecules, whether the final sorting sample will have a greater impact on the reading of DNA molecules, and it is difficult to find structural variations in long DNA molecules.

Round 2

Reviewer 2 Report

The paper can be published.